# Brain-Wide Transgene Expression in Mice by Systemic Injection of Genetically Engineered Exosomes: CAP-Exosomes

**DOI:** 10.3390/ph17030270

**Published:** 2024-02-20

**Authors:** Saumyendra N. Sarkar, Deborah Corbin, James W. Simpkins

**Affiliations:** Department of Neuroscience, Rockefeller Neuroscience Institute, School of Medicine, West Virginia University, 1 Medical Center Drive, 5, Morgantown, WV 265056, USA; sarkarsaumyendra4@gmail.com (S.N.S.); drcorbin@hsc.wvu.edu (D.C.)

**Keywords:** exosomes, CAP-exosomes, blood–brain barrier, gene delivery, neurodegenerative diseases

## Abstract

The bottleneck in drug discovery for central nervous system diseases is the absence of effective systemic drug delivery technology for delivering therapeutic drugs into the brain. Despite the advances in the technology used in drug discovery, such as Adeno-Associated Virus (AAV) vectors, the development of drugs for central nervous system diseases remains challenging. In this manuscript, we describe, for the first time, the development of a workflow to generate a novel brain-targeted drug delivery system that involves the generation of genetically engineered exosomes by first selecting various functional AAV capsid-specific peptides (collectively called CAPs) known to be involved in brain-targeted high-expression gene delivery, and then expressing the CAPs in frame with lysosome-associated membrane glycoprotein (Lamp2b) followed by expressing CAP-Lamp2b fusion protein on the surface of mesenchymal stem cell-derived exosomes, thus generating CAP-exosomes. Intravenous injection of green fluorescent protein (GFP) gene-loaded CAP-exosomes in mice transferred the GFP gene throughout the CNS as measured by monitoring brain sections for GFP expression with confocal microscopy. GFP gene transfer efficiency was at least 20-fold greater than that of the control Lamp2b-exosomes, and GFP gene transduction to mouse liver was low.

## 1. Introduction

The aging of our population is accompanied by an increase in the numbers of patients with neurological disorders, such as Alzheimer’s disease (AD) and Parkinson’s disease (PD). In particular, AD incurs an enormous personal cost to those affected, and the worldwide financial cost in 2010 was estimated at USD 604 billion [1]. It therefore represents a major and rising public health concern, and there is an urgent need to develop more effective therapies to treat and delay the onset of this disease.

Despite the advances in the technology used in drug discovery, the development of drugs for central nervous system diseases remains challenging [2,3]. The failure rate for new drugs targeting important central nervous system diseases is high compared to most other areas of drug discovery. The main reason for failure is the poor penetration efficacy across the blood–brain barrier (BBB). The BBB is formed by the brain capillary endothelium in the central nervous system and prevents the brain uptake of 80% of circulating small molecule drugs and 100% of large molecules such as protein therapeutics, RNAi drugs, and other therapeutic genes [4]. The only way drugs or genes can be distributed widely in the brain is through the transvascular route following injection into the blood stream. However, this transvascular route requires that the drug has the ability to undergo transport across the BBB. Molecular Trojan horses are genetically engineered proteins that cross the BBB via brain endothelial cell receptor-mediated transport processes, which provide a brain drug targeting technology that allows for the non-invasive delivery of large molecule therapeutics to the brain. The development of BBB drug targeting technologies is an underdeveloped area of drug discovery for CNS diseases. In preclinical studies, a molecular Trojan horse, TfRMAb-EPO fusion protein, was administered to AD transgenic mice by IP injections. This treatment caused a reduction in synaptic loss, which correlated with a decrease in microglial activation and an increase in spatial memory [5]. In PD, there is a progressive deterioration of dopamine (DA) neurons, and therapeutic strategies are aimed at preventing DA loss. However, lack of effective brain delivery approaches limits this strategy. In one study, a molecular Trojan horse was used for substantia nigra-targeted delivery of a BBB-penetrating peptide (RVG29) conjugated to the surface of nanoparticles loaded with the natural autophagy inducer 4,4′-dimethoxychalcone (DMC) (designated as RVG-nDMC). It has been demonstrated that RVG-nDMC ameliorates motor deficits and nigral DA neuronal death in PD mice [6].

Nanotechnologies such as viral and non-viral vectors allow for the development of brain-targeted gene delivery systems. In the case of AAV nanoparticles, it was discovered that certain AAV serotypes, e.g., AAV9, undergo transvascular transport across the BBB following an IV injection [7]. AAV gene therapy of the spinal cord is now FDA-approved for the treatment of infantile Spinal Muscular Atrophy (SMA) with a single IV dose of 2 × 10^14^ vector genome per kg (vg/kg) of the scAAV9 encoding the Survival Motor Neuron 1 (SMN1) gene [8]. This gene therapeutic is designated as scAAV9.CB.hSMN (Zolgensma^@^, Switzerland). The FDA-approved IV dose is associated with toxicity in juvenile primates, including elevations of liver enzymes, liver failure, degeneration of dorsal root ganglia, proprioceptive deficits, and ataxia [9]. Zolgensma^@^ IV AAV gene therapy is approved for one-time treatment, because subsequent doses of AAV may cause a potentially severe immune reaction due to the immunogenicity of the AAV capsid protein [10]. Therefore, the current AAV vector requires improvements in transduction potency, antibody evasion and cell/tissue specificity to allow for the use of lower and safer vector doses. Recently, an important capsid variant of AAV9 was generated by sequential engineering of two capsid surface-exposed loops in the capsid protein, VP3. AAV particles with this variant of capsid enabled brain-wide transgene expression and decreased liver targeting after intravenous delivery in mice and marmosets [11]. More specifically, seven amino acid insertion-specific surface-exposed loops in PHPeB virus resulted in increased BBB crossing and delivery of gene. It has been shown that increased brain tropism of PHPeB is due to high-affinity binding of its surface-exposed loop with brain endothelial receptors [12]. Furthermore, a new AAV particle was generated with the above in [11]. Recent studies, by various groups, using a directed evolution strategy in mice and macaques have attempted to generate genetically engineered AAV capsids that enable increased BBB crossing, increased tropism for the brain and reduced tropism for other peripheral organs after systemic administration of viruses with newly generated capsids in both mice and non-human primates (NHPs) [13,14,15]. Also, it has been reported that CNS-tropic behavior of certain AAV9 capsid variant peptides are transferable to other AAV serotypes [16].

In addition to AAV vectors, exosomes, a heterogeneous group of nano-sized natural membrane vesicles, are being developed as non-viral vectors for drug delivery to the brain since they can overcome the BBB penetrance issues. However, the low targeting ability and size-dependent cellular uptake of native exosomes could profoundly affect their delivery performance. Using engineering technology, exosomes can obtain active targeting ability to accumulate in specific cell types and tissues by attaching targeting units to the membrane surface or loading them into cavities. Recent studies have indicated that brain-targeted exosomes from mesenchymal stem cell (MSC) can be created by genetically engineering the exosomes with rabies virus glycoprotein (RVG) peptide expressing Lamp2b, an exosomal membrane protein. However, RVG exosomes also transduce cells in the liver [13].

We hypothesized that by displaying these two capsid loops in the outer membrane surface of exosomes it could gain transgene-loaded exosome’s functional ability to interact with brain capillary endothelial cell receptors followed by crossing BBB and allow brain-wide transgene expression without liver targeting. To this end, we engineered exosomes by first designing nucleotide sequences to encode nuclear localization signal (NLS) signal from AAV9 VP1 capsid protein, endosomal escape signal, AAV9 VP1 capsid protein loop1, spacer, and loop2 peptides to introduce the targeting ligands at the restriction site in the N-terminus of exosomal protein, Lamp2b.

## 2. Results

### 2.1. Characterization of Genetically Engineered CAP-Lamp2b-Hemagglutinin (HA)-Exosomes and Lamp2b-Exosomes

We developed a workflow to generate a novel brain-targeted drug delivery system (DDS) that involves generation of genetically engineered exosomes by first selecting various functional AAV capsid-specific peptides (collectively called CAPs) known to be involved in brain-targeted high-expression gene delivery, and then expressing the CAPs in frame with lysosome-associated membrane glycoprotein (Lamp2b) followed by expressing CAP-Lamp2b fusion protein on the surface of mesenchymal stem cell-derived exosomes, thus generating CAP-exosomes. (Figure 1A). For this study, a recombinant plasmid expression vector was constructed by designing nucleotide sequences to encode the targeting ligand, CAP domain, which consists of nuclear localization signal (NLS) from AAV9 capsid protein VP1 [17], endosomal escaping signal from VP1 [18], heptamer amino acid insertion sequences in variable region VIII of AAV9 [17], spacer amino acids, and heptamer amino acid substitution in variable region IV of AAVCAPB22 [19]. CAP domain-specific forward and reverse-stranded DNA oligos were synthesized and obtained from Integrated DNA Technology (IDT, USA). After annealing the oligos, double-stranded CAP domain was ligated at the restriction site BsaB1 in frame with the N-terminus of exosomal protein Lamp2b in pcDNA-GNSTM-3-FLAG10-Lamp2b-HA plasmid (Hemagglutinin, HA). The pcDNA-GNSTM-3-FLAG10-CAP-Lamp2b-HA-generated plasmid (Appendix A) in this study and control plasmid pcDNAHygro-Lamp2b (Addgene #86029) were used for the electroporation of bone marrow-derived mesenchymal stem cells (BM-MSCs) to produce CAP-Lamp2b fusion protein and Lamp2b-positive MSC-derived genetically engineered exosomes. To validate whether the CAP-lamp2b-HA and the Hygro-Lamp2b plasmids were successfully electroporated into BM-MSCs, and in case of pcDNA-GNSTM-3-FLAG10-CAP-Lamp2b-HA plasmid, the expression of CAP-Lamp2b-HA fusion protein, we assessed the protein levels of Lamp2b and CAP-Lamp2b-HA fusion protein in BM-MSCs 48 h after electroporation. Compared with BM-MSCs electroporated by pcDNA-Hygro-Lamp2b plasmid, the Western blotting showed that the AAV-CAP-Lamp2b-HA-transfected cells expressed HA-Lamp2b fusion protein (Figure 1B, compared to Lane 1 in both the right and left diagrams); if the CAP domain was not cloned at the N-terminal in frame with Lamp2b, then HA expression will be out of frame and, as a result, both Lamp2b and HA antibody-specific signal could not be detected in the same protein by Western blotting. Next, we made large-scale tissue culture growth of BM-MSCs and electroporation with the CAP-Lamp2b-HA and Hygro-Lamp2b plasmid DNA; from the culture supernatant of BM-MSCs, respective exosomes were purified by ultracentrifugation. Western blot analysis of a total of 10^9^ exosomal particles (average size of 100–150 nm) showed that both Lamp2b and HA-specific signals are present, as shown in Figure 1C (Lane 2 indicates the CAP-Lamp2b-HA fusion protein inclusion into the exosomes).

### 2.2. Targeting Efficiency and Specificity of CAP-Exosomes and Lamp2b-Exosomes in the Brain

To investigate the potential of CAP-exosomes to specifically deliver a transgene, (GFP gene expression plasmid), we have developed a workflow, as shown in Figure 2A. First, CAP-exosomes and Lamp2b-exosomes were labeled with 1,1′-dioctadecyl-3,3,3′,3′-tetramethylindocarbocyanine perchlorate (DiI) dye. To wash away free DiI, exosomes were resuspended in 1× phosphate- buffered saline (1XPBS) and pelleted by ultracentrifugation. Next, pelleted DiI-labeled exosomes were resuspended in electroporation buffer, and a total of 50 μg of exosomes corresponding to ~5 × 109 particles were mixed with 50 μg of GFP expression plasmid DNA in a final volume of 100 μL electroporation buffer for a single IV injection to be administered into FVB mice. GFP expression plasmid was inserted into respective exosomes using 4-D-Nucleofector (instruments and reagents from Lonza, Basel, Switzerland) following the company’s instructions. Mice were sacrificed 10 days after IV injection, and serial coronal sections were made and images were taken with a confocal microscope.

In order to detect the presence of the exosomes derived from hygro-Lamp2b plasmid-transfected MSCs in the brain and whether CAP-exosomes modification enhanced BBB crossing and entry into the brain cells, the slides of brain sections were observed under fluorescence microscope on day 5 after injection. The DiI-labeled exosomes were found in the frontal cortex of both the MSC-Lamp2b-Exo group (Figure 2C) and the MSC-CAP-HA-Exo group. But, when further compared to the number of DAPI-stained nuclei and DiI-stained exosomes in red fluorescence, more DiI-labeled exosomes were found in the FC of the mice in the MSC-CAP HA-Exo (Figure 2B) group than that in the MSC-Lamp2b-Exo group (Figure 2C).

### 2.3. Systemic Delivery of CAP-Exosomes Loaded GFP-Gene Resulted in Brain-Wide GFP Expression and Decreased Liver Targeting 

In order to determine the CNS and liver-targeting efficiency of CAP-exosomes, various brain regions and liver sections were made from the same mice sacrificed 10 days after IV injection of DiI-labeled and GFP-loaded exosomes and analyzed by confocal microscopy. In the frontal cortex (FC) of the mice, increased number of DiI-labeled exosomes were found (3D confocal micrographs, Figure 2D, red fluorescence) along with high-efficiency GFP transgene expression (Figure 2E, green fluorescence). Further analysis by visual inspection of merged (Red + Green) confocal micrographs (Figure 2F) revealed that a very high percentage of exosomes was either not loaded with GFP plasmid during electroporation or GFP-loaded exosomes were unable to escape endosomes and failed to transfer the GFP gene to the nucleus of brain cells for transcription. In order to determine the capabilities of CAP-exosomes for neuron-specific brain-wide gene transfer, brain sections from the same mice used for the analysis shown in Figure 2D were stained with neuronal marker NeuN antibody, and the whole hippocampus regions were analyzed with confocal microscopy using low-power (10×) objective lens. Comparative studies of NeuN-stained micrographs (as shown in H) with DiI-labeled exosomes (2H, red) and GFP expression (Figure 2I, green) indicate that systemically delivered CAP-exosome-loaded GFP transgene is capable of brain-wide and possibly neuron-specific delivery of genes, as well as being expressed in the mouse brain.

For further studies to determine whether the high expression of intravenously delivered CAP-exosome-loaded GFP gene in CNS cells while minimizing off-target expression, liver sections obtained from the same mice, used for the analysis shown in Figure 2, were analyzed by confocal microscopy. Levels of both red (Figure 2J) and green (2K) fluorescence were low, indicating that the CAP domain expressing exosomes minimizes off-target GFP gene expression. In a separate sets of animals, IV injection of DiI-labeled and GFP-loaded hygro-Lamp2b exosomes into mice resulted in high exosomal uptake (Figure 2L) and GFP gene expression (Figure 2M) in the animal’s liver.

To further characterize brain-specific GFP gene delivery by CAP-exosomes compared with the control Lamp2b-exosomes, we stained neurons with NeuN and used DAPI nuclei staining for liver cells and quantified the gene transfer efficiency and target specificity of each exosome for each cell type across various brain regions and in the liver (Figure 3A,B). Quantification of the total number of neuron-specific NeuN-stained cells expressing GFP in the hippocampus (CA3-region) shows that 9% of neurons were expressing GFP in case of CAP-exosomes injected mice versus 1.5% in the case of control exosomes. In the cortical region of mice injected with CAP-exosomes, 20% of neurons were GFP-positive compared with 2% in the case of control exosomes. In contrast, neurons were expressed in the GFP gene with high efficiency in CAP-exosome-injected mice compared with control exosomes; GFP expression in liver cells was 5–6-fold lower in CAP-exosome-treated mice compared with control exosomes. This indicates that the CAP-exosomes have brain specificity in mice. Scale bars for the B, C, D, E and F 3D confocal micrographs are shown in white rectangles with values of X = 125 μm, Y = 135 μm, and Z = 20 μm, and for J, K, L, and M, 10 μm white scale bars are shown at the bottom right corner of these micrographs. Scale bars for the B, C, D, E and F 3D confocal micrographs are shown in white rectangles with values of X = 125 μm, Y = 135 μm, and Z = 20 μm, and for J, K, L, and M, 10 μm white scale bars are shown at the bottom right corner of these micrographs.

## 3. Discussion

The goal of this study was to genetically engineer exosomes to confer them the ability to cross the BBB efficiently and then further refine the exosomes for the delivery of drugs to neurons. To achieve this specific goal, we first chose two variants of AAV9 capsid surface-exposed loop-specific peptide along with NLS and endosomal escaping signal peptides as the ligand (we called this peptide as CAP domain) and fused with transmembrane protein Lamp2b that is expressed on the surface of the exosomes. Subsequently, BM-MSCs were chosen as donor cells and transfected with plasmids encoding the CAP-Lamp2b fusion proteins or the engineered exosomes, called CAP-exosomes, bearing the targeting ligands on their surface membrane. We chose CAP domain as the ligand because it has been reported that the surface-exposed peptide loop structure is responsible for conferring AAV’s the ability to cross BBB efficiently, and this variant capsid also enable brain-wide transgene expression and decreased liver targeting after intravenous delivery in mice and marmosets [11].

Intravenous injection of green fluorescent protein (GFP) gene-loaded CAP-exosomes in mice transfer the GFP gene throughout the CNS as measured by monitoring brain sections (hippocampus and frontal cortex) for GFP expression with confocal microscopy. GFP gene transfer efficiency was at least 20-fold greater than that of control Lamp2b-exosomes. GFP gene transduction to mouse liver was low. Also, we chose an expression vector that is capable of producing glycosylation-stabilized CAP domain-Lamp2b fusion protein and is important for increasing the expression of CAP domain on exosomes [20]. The strategy used in this study for generating CAP-exosomes may be particularly useful for translating promising exosome-based drug delivery from preclinical investigation to human trials for brain diseases. The mechanism by which CAP-exosomes increased transport across the blood–brain barrier is not known. It has been reported that genetically engineered surface-exposed loop of the capsids of PHPeB, AAV-B10, and AAV-B22 increased neuronal transduction throughout the brains of mice and marmosets compared to AAV9 [11]. This increased efficiency is believed to result from a novel interaction between virus and the brain endothelial cell receptor LY6A [21]. Although similar 7-mer amino acid substitution and 7-mer insertions were used in CAP-exosomes, further research is needed to identify whether or not CAP-exosomes interact with LY6A. Further research will be needed for success in clinical trials by translating CAP-exosome-mediated drug delivery from mice to primate to human’s brain.

Exosomes have gained attention as cell-derived nanoparticles for treating CNS diseases owing to their potential broad surface engineering capability. Receptor-mediated transcytosis (RMT) exosomes expressing ligands such as LDLR-targeting apolipoprotein B has shown promising results [22]. Although surface-modified CAP-exosomes possessing brain targetability have shown enhanced CNS delivery in preclinical studies described here, the successful development of clinically approved CAP-exosome therapeutics for CNS diseases requires the establishment of methods for monitoring exosomal delivery to the brain parenchyma of mouse to non-human primate (NHP) to human, as well as an elucidation of the mechanisms underlying the BBB crossing of surface-modified exosomes. In future research, biodistribution as well as cellular receptors, including brain endothelial cell receptor binding affinity of CAP-exosomes, will be determined by a single Positron Emission Tomography (PET) in live animal as it has recently been shown that the PHP.eB capsid’s brain endothelial receptor affinity is nearly 20-fold greater than that of AAV9 [12].

## 4. Materials and Methods

### 4.1. Animals and Cell Culture

All rodent procedures were approved by the Institutional Animal Use and Care Committee of West Virginia University.

#### 4.1.1. FVB Mice Bone Marrow Cell Collection

Cell Collection: The anesthetized male and female FVB mice (2–3 months old) were placed in a 100 mm culture dish and washed with 70% ethanol. Tibias and femurs were dissected; muscle, ligaments, and tendons were removed. Next, with micro dissecting scissors, the two ends were excised and a needle with 2 × 5 ml syringe, filled with sterile phosphate-buffered saline, was inserted into the bone cavity and used to slowly flush the marrow out from the total volume of 10 mL PBS/mice used for extracting bone marrows from two of each tibias, femurs, and humeri into a culture dish containing BM-MSC growth medium. BM-MSC colony growth was expanded according to the protocol described by Soleimani and Nadri [23]. 

#### 4.1.2. Plasmid DNA Transfection in BM-MSC Cells and Loading into Exosomes by Electroporation

BM-MSCs were used for transfection of CAP-Lamp2b, and hygro-Lamp2b expression cassette containing plasmid DNA by electroporation. GFP expression plasmid was inserted into CAP-exosomes using 4-D-Nucleofector (instruments Core Unit: AAF-1001B and reagents from Lonza, Basel, Switzerland, following the company’s instructions). In brief, a P3 primary cell 4D Nucleo factor kit (V4XP3024, catalog no, Lonza) was used first for standardizing the efficiency of plasmid DNA transfection in cells and loading in exosomes. Using 5 × 10^6^ cells, 5 μg DNA, and 7 different electroporation program available in the 4D Nucleofactor X unit, highest efficiency was achieved when using program No.1. In the case of both exosomes, 1 × 10^9^ particles, 10 μg of GFP-plasmid DNA, 100 μL buffer, and the use of 7 different electroporation program, highest efficiency was achieved by program No. 5, as determined by measuring GFP gene expression in BM-MSCs after transfection with GFP gene-loaded exosomes.

### 4.2. Construction of CAP-Lamp2b Expression Vector

Nucleotide sequences were designed to encode the targeting ligand, named CAP domain, which consists of NLS signal from AAV9 capsid protein VP1, endosomal escaping signal from VP1, heptamer amino acid insertion sequences in variable region VIII of AAV9, spacer amino acids, and heptamer amino acid substitution in variable region IV of AAVCAPB22. CAP domain-specific forward and reverse-stranded DNA oligos were synthesized and obtained from Integrated DNA Technology (IDT, Coralville, IA, USA). After annealing the oligos, double-stranded CAP domain was ligated at the restriction site BsaB1 in frame with the N-terminus of the exosomal protein Lamp2b in pcDNA-GNSTM-3-FLAG10-Lamp2b-HA-plasmid (Addgene#71293). The pcDNA-GNSTM-3-FLAG10-CAP-Lamp2b-HA-generated plasmid in this study and control plasmid, pcDNAHygro-Lamp2b (Addgene #86029), were used for electroporation of bone marrow-derived mesenchymal stem cells (BM-MSCs) to produce CAP-Lamp2b fusion protein and Lamp2b-positive MSC-derived genetically engineered exosomes. Nucleotide sequences encoding variant AAV9 capsid surface-exposed peptide loops used in this study are shown in Figure 4.

### 4.3. Isolation of Exosomes

From CAP-Lamp2b or hygro-Lamp2b plasmid DNA-transfected BM-MSCs, cell culture-conditioned media exosomes were isolated and purified either by differential centrifugation as described by Thery et al. in [18] or by using the ExoEasy Maxi Kit (Qiagen, Venlo, The Netherlands). Final ultracentrifugation was carried out at 100,000× *g* using Beckman Model L8-70MR ultracentrifuge.

### 4.4. Size and Concentration Determination of Exosomes

The sizes and concentrations of purified and DiI-labeled exosomes were determined by using a Malvern Panalytical NanoSight instrument (Malvern, UK) according to the company’s instructions.

### 4.5. Western Blotting

For Western blotting, BM-MSCs electroporated with pcDNA-hygro-Lamp2b or CAP-Lamp2b-HA-plasmid were lysed by RIPA buffer. Polyacrylamide gel electrophoresis was run using Mini-PROTEAN Tetra Vertical Electrophoresis Cell, and proteins were transferred to PDF nylon membrane. Blots were incubated with primary antibodies overnight at 4 °C. The primary antibodies used were as follows: goat anti-Lamp2b (1:1000, Abcam (Cambridge, MA, USA) and rabbit anti-HA antibody (Cell Signaling Technology). Corresponding IR Dye secondary Donkey anti-goat and Donkey anti-rabbit (1:20,000, LiCOR (Lincoln, NE, USA)) were incubated for 1 h at room temperature. Bands were visualized using Licor imaging system (Odyssey^®^ DLxImaging System (Lincoln, NE, USA).

### 4.6. Tissue Preparation and Immunofluorescence

Mice were deeply anesthetized in an isoflurane box and transcardially perfused with ice-cold 1× PBS and then freshly prepared ice-cold 4% paraformaldehyde (PFA) in 1× PBS. All organs were excised and post-fixed in 15% sucrose containing 4% PFA solution for overnight fixation at 4 °C, followed by another overnight fixation in 30% sucrose. Cryoprotected brains were mounted onto a freezing microtome (model HM 450, Micron, Walldorf, Germany) and sectioned on a coronal plane at 40 μm thickness. In the case of the liver, immediately after perfusing the mice, the liver lobes were removed and cut into 2–3 mm blocks and fixed in 4% paraformaldehyde for 24 h before being placed in 30% sucrose for cryoprotection. The cryoprotected liver blocks were mounted onto a freezing microtome (model HM 450, Micron. Walldorf, Germany) and sectioned at 40 μm thickness.

Serial coronal sections of 40 micron in thickness were prepared on a cryostat. After being incubated with 0.3% Triton X-100 and 3% bovine serum albumin (BSA) in PBS for 1 h, the following primary antibody was incubated overnight at room temperature: rabbit anti-NeuN (1:200, Abcam 177487). The corresponding secondary antibody, Alexa Fluor 647 (1:200, Thermo Fisher Scientific (Waltham, MA, USA), A32795), was incubated for 3 h at room temperature. Cellular nuclei were stained by Hoechst 33342 (1:100, Sigma (Livonia, MI, USA)).

### 4.7. Imaging and Quantification

All GFP green-expressing tissues (liver and brain), presence of DiI (red)-labeled exosomes in the tissues and NeuN-immunostained neurons in brain sections were imaged on a NICON A1R confocal laser microscope using 10× and 40× objective, with matched laser powers, gains and gamma across all samples of the same tissue section. In all cases in which fluorescence intensity was compared between samples, exposure settings and changes to gamma or contrast were maintained across images. The acquired images were processed using NIS-Element imaging software. All CAP-exosome-GFP and Lamp2b-GFP-expressing tissues were imaged on a Zeiss LSM confocal microscope using a 10× objective, with matched laser powers, gains, and gamma across all samples of the same tissue. Visualizations of brain sections were also carried out using Nikon A1R Confocal/N-SIM-E*and/or Zeiss 710 Confocal w/Airyscan. The acquired images were processed in Zen Blue 2 (Zeiss, Okerkochen, Germany)). Processing of all czi images from confocal microscope was performed with FIJI (ImageJ). Colocalization between the GFP signal and NeuN antibody or DAPI staining was performed using FIJI with the cell count automated plugin. The details method used for the quantitation was followed according to the published method [24]. In brief, quantitation of images was carried out in Fiji software, available online, using the following sequential steps:

i. The fluorescent images shown in Figure 3 in a czi file format (Zeiss confocal microscopes) were opened by clicking the ‘File’ tab, and then clicking ‘Open’ to open the file for quantification. A window named ‘Bio-Format Import Options’ pops up.

ii. We used the ‘Hyperstack’ and ‘Colorized’ options as they allow for the independent analysis of each of the fluorescent channels collected in the original experimental data presented here, which were collected using three-color RGB imaging. We clicked on the ‘Split channels’ option that shows the three color channels in three separate windows for the quantitation of each fluorescent channel separately. Finally, we clicked ‘OK’ at the bottom right of the window to proceed to the next step of quantification.

iii. Control and experimental images were analyzed as follows: Selecting Image > Type > 8 bit. By clicking Edit > Selection > Specify to define a region, i.e., ROI. The image with ROI was opened; then, the histogram tool (Analyze > Histogram) was opened and “list” was selected to obtain pixel counts, and the number of Values, both in red and green channels, were recorded. The percentage of GFP gene-loaded CAP-exosomes was calculated by dividing the number of green pixels by the total number of red pixels, multiplied by 100.

### 4.8. Statistics

GraphPad Prism 9 was used for statistical analysis and data representation. All experimental groups were *n* = 4. For the statistical analyses and related graphs, a single data point was defined as two tissue sections per animal, with multiple technical replicates per section when possible. Technical replicates were defined as multiple fields of view per section, with the following numbers for each region or tissue of interest: cortex = 3 hippocampus = 4, liver = 4. Statistical analyses of the data were performed using Prism 9 and the multiple unpaired (non- parametric) *t*-test.

## 5. Conclusions

This study provides a novel strategy for targeted delivery of genetically engineered exosomes, named CAP-exosomes, to the brain. CAP-exosomes’ ability to cross the blood–brain barrier with high expression of delivered gene to the brain cell in rodents empowers new directions for the research and development of CNS drugs and therapeutic possibilities unattainable with AAV gene therapy.

## Figures and Tables

**Figure 1 pharmaceuticals-17-00270-f001:**
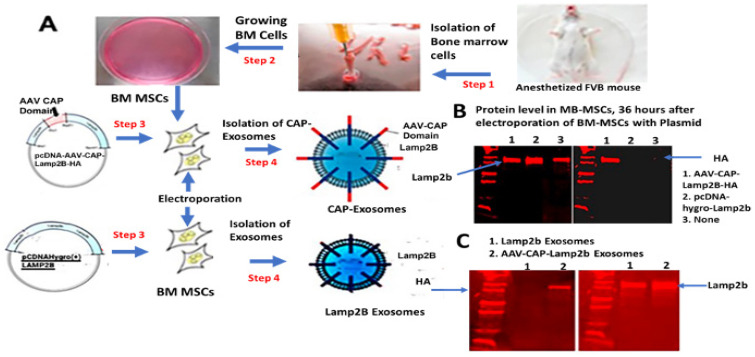
Generation and characterization of genetically engineered MSCs-derived exosomes. (**A**). The schematic diagram showing the various steps including isolation and growth of BM-MSCs derived-AAV-CAP domain-Lamp2b fusion protein expression plasmid recombination, generation of BM-MSC derived CAP-Exosomes and Lamp2b-exosomes. (**B**). Western blot analysis shows expression of fusion protein CAP-Lamp2b-HA in total protein isolated 48 h after electroporation of CAP-Lamp2b with BM-MSCs as both the Hemagglutinin HA-tag antibody specific and Lamp2b- specific positive signal shows in the 1st western blot run ((**B**). Right panel in lane1) and after striping of 2nd western blot ((**B**), Left panel, lane1) respectively, and neither in the protein isolated from pcDNA-hygro-Lamp2b electroporated BM-MSCs ((**B**), Lane 2, Right and Left panel), nor in the total protein isolated from just BM-MSCs ((**B**), Lane 3, Right and Left panel). (**C**). Western blot analysis of a total of 10^9^ exosomal particles (average size of 100–150 nm) showed that both Lamp2b and HA specific signal are present in (**C**) (Lane 2) indicating the CAP-Lamp2b-HA fusion protein inclusion on the exosomes.

**Figure 2 pharmaceuticals-17-00270-f002:**
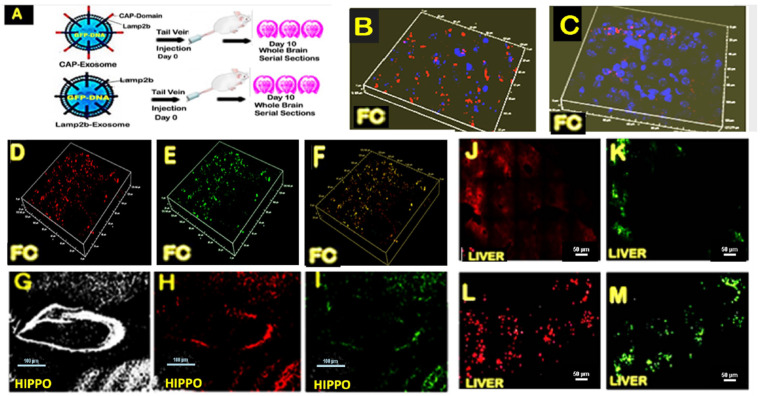
Targeting efficiency and specificity of CAP-Exosomes and Lamp2b-exosomes after systemic delivery. (**A**) a schematic for the production of CAP-exosomes and Lamp-2b-exosomes. Presence of high numbers of DiI-labeled (red) CAP-exosomes (**B**) compared to Lamp2b-exosmes (**C**) in the frontal cortex region of respective mice. In the frontal cortex (FC) of the mice very high DiI-labeled exosomes were found (3D-confocal micrograph (**D**), red fluorescence) along with high efficiency GFP-transgene expression ((**E**), green fluorescence). Note that visual inspection of merging (Red + Green) confocal micrograph (**F**) revealed a high percentage of only red fluorescence but not merged colors. NeuN stained micrograph of the hippocampus ((**G**), white) with DiI labeled exosomes ((**H**), red) and GFP- expression ((**I**), green) indicate that systemically delivered CAP-Exosomes loaded GFP- transgene capable of brain wide and highly neuron specific delivery of genes and its expression in the mouse brain. Both the red (**J**) and green (**K**) fluorescence were low indicating CAP- domain expressing exosomes minimizing off-target GFP-gene expression. In a separate sets of animals IV injection of DiI labeled and GFP loaded hygro-Lamp2b exosomes to mice resulted in high exosomal uptake (**L**) and GFP-gene expression (**M**) in the animal’s liver.

**Figure 3 pharmaceuticals-17-00270-f003:**
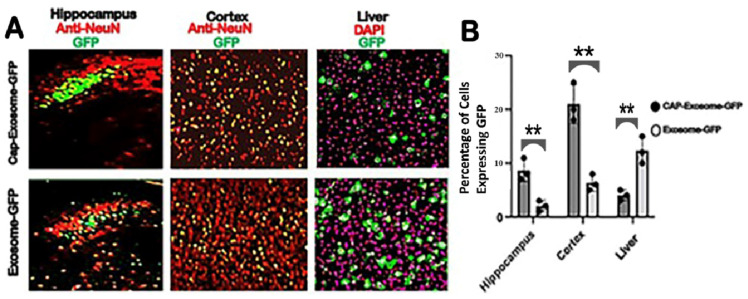
CAP-Exosome-GFP expression is preferred in brain with a significant decrease in liver. CAP-Exosome-GFP and (Lampb2 control) Exosome-GFP (total of 5 μg in 0.1 mL saline buffer/mice) particles were intravenously injected into FVB adult mice. GFP fluorescence was assessed after 10 days of expression. Expression of the total number of Cells expressing GFP and NeuN immune stained shown in (**A**) and percentage of cells expressing GFP shown in (**B**) In the Hippocampus (*n* = 4 animals, ** *p* = 0.0074, control Exosome versus CAP-Exosomes), Cortex (*n* = 4 animals, ** *p* = 0.002, control Lampb2 Exosome Versus CAP-Exosomes). In case of liver total number of cells expressing GFP and DAPI nuclear stained cells shown in (**A**) and percentage of cells expressing GFP shown in (**B**). *n* = 4 animals, ** *p* = 0.005 for control Exosome versus CAP-Exosomes.

**Figure 4 pharmaceuticals-17-00270-f004:**
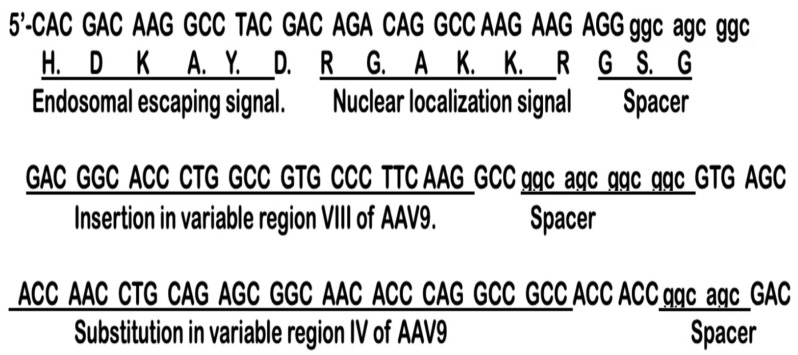
Nucleotide sequences encoding variant AAV9 capsid surface-exposed peptide loops used.

## Data Availability

Original data can be obtained by contacting the corresponding author.

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
