# Peer review of "Brain-Wide Transgene Expression in Mice by Systemic Injection of Genetically Engineered Exosomes: CAP-Exosomes"

_pharmaceuticals, 2024, doi:10.3390/ph17030270_

Round 1
Reviewer 1 Report
Comments and Suggestions for Authors
I reviewed the brief report entitled “Mouse Brain-Wide Transgene Expression by Systemic Injection of Genetically Engineered Exosomes: CAP-Exosomes”, submitted by Sarkar et al. for publication in Pharmaceuticals. The manuscript is well written and summarizes the content and research in a comprehensive way. I regard this study as a valuable contribution in the field of genetically engineered exosomes after addressing some major comments.
1. I would suggest that authors explain the aim of this article and how does this article differ what is already out there.
2. The literature in the comparison with the genetically engineered proteins (Molecular Trojan horses) that cross the BBB via brain endothelial cell receptor-mediated transport processes and provide a brain drug targeting technology that allows for the non-invasive delivery of large molecule therapeutics to the brain is not comprehensive enough. Therefore, the authors should add some more literature.
3. Authors can incorporate a graphical figure to get the basic concept.
4. The manuscript should be checked for possible typos i.e. line 49,130 etc
5. Specify the details of all the instruments (model, manufacturer etc.) used in the manuscript.
6. In section 2.1, mention the quantity of phosphate buffer saline to be filled in syringe used to slowly flush the marrow out into a culture dish containing BM-MSC growth medium.
7. The degree symbol is missing from the line 179.
8. The extant literature review is deemed unsatisfactory in its present state due to the utilization of outdated references.
9. Reference section needs improvement. The report lacks citation of recent works, necessitating the inclusion of more contemporary research, notably the absence of references from the year 2023.
10. The articulation of sentences requires modification, and comprehensive formatting adjustments are warranted in the manuscript.
11. The quality of figures within the results section is suboptimal; thus, inclusion of high-resolution images is imperative.
12. Additionally, the author is encouraged to incorporate a conclusion section to succinctly encapsulate and summarize their findings.
Comments on the Quality of English Language
Language needs improvement. Please work through the MS carefully from this perspective.
Author Response
Reviewer 1 We thank Reviewer 1 for his/her insightful comments on our original manuscript. Our responses and changes in the revised manuscript are indicated below and the changes in the revised manuscript are indicated in blue text. 1. I would suggest that authors explain the aim of this article and how does this article differ what is already out there. This is now indicated in the revised abstract. 2. The literature in the comparison with the genetically engineered proteins (Molecular Trojan horses) that cross the BBB via brain endothelial cell receptor-mediated transport processes and provide a brain drug targeting technology that allows for the non-invasive delivery of large molecule therapeutics to the brain is not comprehensive enough. Therefore, the authors should add some more literature. More references are added. 3. Authors can incorporate a graphical figure to get the basic concept. A Graphical Figure is now included in the revised manuscript. 4. The manuscript should be checked for possible typos i.e. line 49,130 etc We have thoroughly reviewed the revised manuscript for typos. 3. Specify the details of all the instruments (model, manufacturer etc.) used in the manuscript. Details of all the instruments are now mentioned in the method section. 6. In section 2.1, mention the quantity of phosphate buffer saline to be filled in syringe used to slowly flush the marrow out into a culture dish containing BM-MSC growth medium. It is now specified in the Method section the total volume of 10 ml PBS /mice used for extracting bone marrows from two of each tibias, femurs, and humeri. 7. The degree symbol is missing from the line 179. This correction is now made. 8. The extant literature review is deemed unsatisfactory in its present state due to the utilization of outdated references. References covering year 2023 are now included 9. Reference section needs improvement. The report lacks citation of recent works, necessitating the inclusion of more contemporary research, notably the absence of references from the year 2023. References covering year 2023 are now included. 10. The articulation of sentences requires modification, and comprehensive formatting adjustments are warranted in the manuscript. We have thoroughly reviewed the revised manuscript for these errors. 11. The quality of figures within the results section is suboptimal; thus, inclusion of high-resolution images is imperative. All Low-Res figures are now converted 70 High-Res (600) in Photoshop. 12. Additionally, the author is encouraged to incorporate a conclusion section to succinctly encapsulate and summarize their findings. A conclusion has now been added to the revised manuscript.

Reviewer 2 Report
Comments and Suggestions for Authors
The manuscript Mouse brain-wide transgene expression by systemic injection of genetically engineered exosomes: CAP-Exosomes' is a very interesting paper which explain Mouse brain-wide transgene expression by systemic injection of genetically engineered exosomes: CAP-Exosomes'. The report provides a large amount of data.
The report is good and complete and the different sections of the report are well balanced and adequately supported by the data provided. The report is appropriate from the aims and scope of the journal and is potentially publishable "Pharmaceuticals".
1. Introduction section is good but n some more material in introduction section particularly form studies publishes reputed international Journals (year 2018-2023) could be added.
2. Hypothesis is missing. Include it in introduction section
3. Figures 1 and 2 can be simplified and break so reader can easily understand them
4. Add some graphs in result section, if appropriate
5. Discussion section can be improved and supported with some more recent citations .
6. The whole article need to check for grammar and typos error
7. All references must be according to journal guidelines
Author Response
Reviewer 2 We thank Reviewer 2 for his/her insightful comments on our original manuscript. Our responses and changes in the revised manuscript are indicated below and the changes in the revised manuscript are indicated in blue text. 1. Introduction section is good but n some more material in introduction section particularly form studies publishes reputed international Journals (year 2018-2023) could be added. More recent and relevant studies related papers are added in the introduction. 2. Hypothesis is missing. Include it in introduction section The hypothesis is now in the final introductory paragraph. 2. Figures 1 and 2 can be simplified and break so reader can easily understand them Breaking may simplify the figures, but the data will be unlinked and therefore less understandable. 3. Add some graphs in result section, if appropriate No other data except that used for the graph in Fig. 3 is available. 6. The whole article need to check for grammar and typos error We have reviewed the manuscript for these errors. 7. All references must be according to journal guidelines All references are now formatted according to the journal guidelines.

Reviewer 3 Report
Comments and Suggestions for Authors
The authors demonstrated brain targeting of genetically modified exosomes as a tool for delivering plasmid DNA. While the study had interesting results, the quality of the manuscript was insufficient.
1. The citation of the literature was insufficient. At least, the original article(s) for the loading of plasmid DNA into exosomes by electroporation should be cited.
2. The abstract section was not well-summarized. It may be a simple copy and paste of the introduction section. The abstract should not contain references (reference numbers). The authors should describe the results and significance of the study in the abstract.
3. About the paragraph line 433-435, what does it mean? I feel oddness in the paragraph as the last part of the manuscript. My feeling may come from the lack of a conclusion.
4. AAVs are DNA viruses. So, the description “virus genome packaged therapeutic DNA (genes) or RNA molecules” may be wrong.
5. The resolution of the figures was too poor to understand (especially Fig. 1). Please improve it.
6. The manuscript lacked centrifugal forces for the experiments of ultracentrifugation.
7. About Figs 1B and 1C, the order of the right and left panels may be inversed. For readability, it is better to unify the order of the panels in Figs 1B and 1C.
8. There may be a lot of typos. I cannot indicate all of the typos.
I. The term “trozan” may be a typo of “Trojan”. And, “T” in “Trojan” should be in large capital.
II. “Zolgensmar” may be a typo of “Zolgensma”. The last “r” in “Zolgensmar” may be a registered trademark “®”.
III. Malvern Panalytical (“Malvin” as indicated in line 174 may be a typo) is a UK company, not a USA one.
IV. “Zen Bue 2” may be a typo of “Zen Blue 2”.
V. “GTP expressing CAP-Exosomes” may be a typo of “GFP-expressing CAP Exosomes”. “GFP gene-loaded CAP Exosomes” may be better.
VI. “ells” in line 362 may be “cells”.
VII. In line 416, “or not” appeared twice.
Author Response
Reviewer 3 We thank Reviewer 3 for his/her insightful comments on our original manuscript. Our responses and changes in the revised manuscript are indicated below and the changes in the revised manuscript are indicated in blue text. 1. The citation of the literature was insufficient. At least, the original article(s) for the loading of plasmid DNA into exosomes by electroporation should be cited. We standardized our electroporation method using Lonza[s nucleofector as described in the method section. Although in the following cited reference, Lonza’s instrument was used but cannot be cited because it is now retracted Z, Yang J, Meng Q. Endothelial progenitor cell-derived exosomes, loaded with miR-126, promoted , but cannot be cited because it I now retracted.deep vein thrombosis resolution and recanalization. Stem Cell Res Ther. 2018 Aug 23;9(1):223. doi: 10.1186/s13287-018-0952-8 2. The abstract section was not well-summarized. It may be a simple copy and paste of the introduction section. The abstract should not contain references (reference numbers). The authors should describe the results and significance of the study in the abstract. The Abstract has been re-written. 4. About the paragraph line 433-435, what does it mean? I feel oddness in the paragraph as the last part of the manuscript. My feeling may come from the lack of a conclusion. We have now added a conclusion to the revised manuscript. 4. AAVs are DNA viruses. So, the description “virus genome packaged therapeutic DNA (genes) or RNA molecules” may be wrong. Reviewer is right; it would be plasmid encoding RNA or micro-RNA. 5. The resolution of the figures was too poor to understand (especially Fig. 1). Please improve it. All the Figures are now in high resolution. 6. The manuscript lacked centrifugal forces for the experiments of ultracentrifugation. This information has been added to the revised manuscript. 7. About Figs 1B and 1C, the order of the right and left panels may be inversed. For readability, it is better to unify the order of the panels in Figs 1B and 1C. The figures are correct. Red is Dil-labeled cells and there are many more in Cap-exosome treated sections. 8. There may be a lot of typos. I cannot indicate all of the typos. All typo have been identified and corrected. 9. The term “trozan” may be a typo of “Trojan”. And, “T” in “Trojan” should be in large capital. These changes have been made in the introduction and by removing reference to “Trojan” horse from the abstract. 10. “Zolgensmar” may be a typo of “Zolgensma”. The last “r” in “Zolgensmar” may be a registered trademark “®”. The suggested change has been made. 11. Malvern Panalytical (“Malvin” as indicated in line 174 may be a typo) is a UK company, not a USA one. The suggested change has been made. 12. “Zen Bue 2” may be a typo of “Zen Blue 2”. This correction has been made. 13. “GTP expressing CAP-Exosomes” may be a typo of “GFP-expressing CAP Exosomes”. “GFP gene-loaded CAP Exosomes” may be better. This correction has been made. 14 . “ells” in line 362 may be “cells”. This correction has been made. 15. In line 416, “or not” appeared twice This correction has been made.

Reviewer 4 Report
Comments and Suggestions for Authors
Authors need to address the following issues before publications.
1. PD is not defined in the abstract even though it is defined in the introduction.
2. Authors need to provide more context discussing the functions and importance of a seven amino acid insertion specific surface exposed loop and a new seven amino acid substitution specific surface exposed loop.
3. Line 109, which two loops are the authors referring to? The two loops associated with AAV9 in the previous paragraph?
4. Figure 1 and figure 3's quality is too low and texts are not legible. Please revise.
5. Figure 2a is not mentioned in the caption.
6. Line 338-341, authors stated that figure is not shown but isn't figure 2f showing the red+green merged color images? Please also revise the discussion from these texts as it is confusing.
7. The authors used inconsistent abbreviations for exosomes. Does the CAP-exosome from figure 2 and the GFP-containing exosomes in figure 3 also contain Lamp2b? If so, please include Lamp2b in their abbreviations to avoid confusions.
8. There are no scale bars in figure 2 and 3. The scale bar information in line 369-375 does not make sense.
9. Texts in line 433-435 seems out of context. Please revise.
10. The abstract failed to introduce exosome which is the main focus of the this work.
11. In order to be more convincing , authors also need to provide relevant pharmacokinetics and toxicity data of the engineered exosomes.
Comments on the Quality of English Language
Some sentences sound like long phrases instead of complete sentences. Please thoroughly check throughout the paper and revise.
Author Response
We thank Reviewer 4 for his/her insightful comments on our original manuscript. Our responses and changes in the revised manuscript are indicated below and the changes in the revised manuscript are indicated in blue text. Reviewer 4 1. PD is not defined in the abstract even though it is defined in the introduction. The abstract has been modified and PD was eliminated from the revised abstract. 2. Authors need to provide more context discussing the functions and importance of a seven amino acid insertion specific surface exposed loop and a new seven amino acid substitution specific surface exposed loop. Line 109, which two loops are the authors referring to? The two loops associated with AAV9 in the previous paragraph? In the preceding paragraph. we described the two loops as engineered 7AA insertion and substitution in the variable region of two surface exposed loops in AAV9 capsid protein The functional role of 7-aminoacid variant (Substitution) into the AA455 loop and 7-amino acid variant (Inertion) into the 588 loop has been implicated in enhancing interaction with brain endothelial cell receptor and detargeting viral transduction from peripheral organ. 3. Two Loop and 7-AA shown here is taken from this version posted June 17, 2020 ; https://doi.org/10.1101/2020.06.16.152975doi: bioRxiv preprint 4. Figure 1 and figure 3's quality is too low and texts are not legible. Please revise. Higher quality figures are now provided. 5. Figure 2a is not mentioned in the caption. Figure 2A is now described in the captions. 6. Line 338-341, authors stated that figure is not shown but isn't figure 2f showing the red+green merged color images? Please also revise the discussion from these texts as it is confusing. Thank you for this comment. This is now changed in the manuscript. 7. The authors used inconsistent abbreviations for exosomes. Does the CAP-exosome from figure 2 and the GFP-containing exosomes in figure 3 also contain Lamp2b? If so, please include Lamp2b in their abbreviations to avoid confusions. Yes, the CAP-exosome in figures 2 and 3 contain Lampb2. This is now indicated in the revised manuscript. 8. There are no scale bars in figure 2 and 3. The scale bar information in line 369-375 does not make sense. . All scale bar are correct and shown here for clarification Please look at white recatangle and the assigned numbers in micrometer scale. Also in case of liver section, right bottom corner white bar shown. Scale bar in cases of B, C, D, E and F 3D confocal micrographs as shown in white rectangle values of X=125m, Y=135m, Z=20m , and in cases of J, K, L, M. 10m white scale bar is shown at the bottom right corner of these micrographs. Scale bar in cases of B, C, D, E and F. 3D confocal micrographs as shown in white rectangle values 9. Texts in line 433-435 seems out of context. Please revise. This sentence has now been removed. 10. The abstract failed to introduce exosome which is the main focus of this work. The abstract has been substantially modified. 11. In order to be more convincing, authors also need to provide relevant pharmacokinetics and toxicity data of the engineered exosomes This recommendation in far beyond the purpose of the current manuscript.

Round 2
Reviewer 1 Report
Comments and Suggestions for Authors
All the comments addressed in the revised manuscript accordingly. The report is accepted in the present form.
Author Response
No response since there were no issues raised.
Reviewer 4 Report
Comments and Suggestions for Authors
All the figures are still in low quality. All the texts are fuzzy and illegible. Figure 1 has unclear dots in the back ground. Scale bars are impossible to be read. Hence, the presentation quality is unacceptable for publication.
Author Response
Both Figure 1 and 2 have been completed revised and are now part of the revised manuscirpt.